To stay or to go: resource diversity alters the dispersal behavior of sympatric cryptic marine nematodes

http://orcid.org/0000-0003-4526-5160 Guden Rodgee Mae 1 rodgeemae.guden@ugent.be
http://orcid.org/0000-0003-3763-6187 Derycke Sofie 1 2
http://orcid.org/0000-0001-6544-9210 Moens Tom 1
1 Marine Biology Unit, Department of Biology, Ghent University , Ghent , Belgium
2 Flanders Research Institute for Agriculture, Fisheries and Food (ILVO), Aquatic Environment and Quality , Oostende , Belgium
Kramer Donald
Electronic publication date: 2025 Jan 13
Publication date: 2025
Volume: 13
Electronic Location ID: e18790
Received 2023 Jun 13; Accepted 2024 Dec 10
Copyright: © 2025 Guden et al.
Copyright year: 2025
Copyright holder: Guden et al.
License: This is an open access article distributed under the terms of the Creative Commons Attribution License, which permits unrestricted use, distribution, reproduction and adaptation in any medium and for any purpose provided that it is properly attributed. For attribution, the original author(s), title, publication source (PeerJ) and either DOI or URL of the article must be cited.
License URL: https://creativecommons.org/licenses/by/4.0/

Keywords: Informed dispersal, Foraging, Patch structure, Cryptic species, Species coexistence

Funding: Ghent University Flemish Science Fund FWO GOH3817N and 01GA1911W FWO EU Horizon 2020 ERA-Net COFUND and ERA Biodiversa 3G0H6816 Belgian Federal Science Policy Belspo BRAIN-be contract BR/175/A1/BIO-Tide-BE Rodgee Mae Guden benefitted from a Ph.D. Scholarship from Ghent University (BOF) during the period of this study. The results of this study were obtained using infrastructure and knowhow provided in the framework of EMBRC Belgium, Flemish Science Fund FWO project GOH3817N. Specific funding for the laboratory experiments was provided by the FWO through project 3G.0192.09, and by the BiodivERsA project BIO-Tide ‘The role of microbial biodiversity in the functioning of marine tidal flat sediments’ (EU Horizon 2020 ERA-Net COFUND), with financial support of FWO (ERA Biodiversa 3G0H6816) and the Belgian Federal Science Policy Belspo (BRAIN-be contract BR/175/A1/BIO-Tide-BE). Additional financial support was granted by the Flemish Science Fund FWO through project 01GA1911W. The funders had no role in study design, data collection and analysis, decision to publish, or preparation of the manuscript.

==============================
Animals can use specific environmental cues to make informed decisions about whether and where to disperse. Patch conditions are known to affect the dispersal behavior of animals, but empirical studies investigating the impact of resource diversity on the dispersal of closely related species are largely lacking. In this study, we investigated how food diversity affects the dispersal behavior of three co-occurring cryptic species of the marine bacterivorous nematode complex Litoditis marina (Pm I, Pm III and Pm IV). Using microcosms composed of a local patch (inoculation plate), a connection tube, and a distant patch (dispersal plate), we examined nematode dispersal patterns with bacteria serving as the food source. Food treatments included low-, medium-, and high-diversity bacterial mixtures of 5, 10, and 15 bacterial strains, respectively. Additionally, a single-strain food resource Escherichia coli was used as a control treatment. Both local and distant patches had either identical food treatments (‘homogeneous patches’) or E. coli in the local patches and more diverse food (low-, medium-, or high-diversity food) in distant patches (‘heterogeneous patches’). Our results show that the dispersal behavior of the cryptic species varies depending on food diversity, indicating that L. marina acquire information about their environment when making dispersal decisions. All three cryptic species tend to disperse faster toward food patches that increase fitness. Pm I and Pm IV exhibited faster dispersal toward patches with a more diverse food source, while Pm III showed similar dispersal rates toward E. coli, medium-diversity, and high-diversity food. This indicates that resource diversity can alter the dispersal behavior of cryptic species and may be an important mechanism to achieve species coexistence in the field.

Introduction

Dispersal is a fundamental ecological process that shapes local and regional diversity patterns, with profound effects on population dynamics, species coexistence, and evolutionary outcomes (Leibold & McPeek, 2006; Ronce, 2007; Bonte et al., 2012). It is essential for most animals to escape unfavorable environmental conditions and/or locate suitable food resources for survival and reproduction (Ronce, 2007; Clobert et al., 2009; Urban et al., 2016). Theoretical and empirical studies demonstrate that dispersal is condition-dependent and an informed process (Clobert et al., 2009). The information for dispersal can be acquired through a variety of cues, such as local population density (Bowler & Benton, 2005; Matthysen, 2005; Bitume et al., 2014), body condition (Bonte & De La Peña, 2009), or abiotic factors (Altermatt & Ebert, 2010). Food availability can also trigger dispersal, with individuals tending to be more dispersive when local resources are scarce (Aguillon & Duckworth, 2015; Fronhofer et al., 2015; Kreuzinger-Janik et al., 2022). Food resources are generally distributed in the environment in patches of different abundances and diversity, but surprisingly little is known on how food diversity influences the dispersal behavior of organisms.

The effects of food diversity in shaping the structure of ecological communities remain a matter of debate (Duffy, 2002). Some theoretical predictions indicate that diverse food sources will be less beneficial for a population of consumers because these are more likely to contain food that is resistant to consumption (‘variance-in-edibility hypothesis’) (Leibold, 1989; Hillebrand & Shurin, 2005) or because they can reduce both the relative and absolute abundances of the preferred food of consumers (‘dilution or resource concentration hypothesis’) (Andow, 1991; Joshi et al., 2004; Keesing, Holt & Ostfeld, 2006). Alternatively, a more diverse food source may be advantageous for a population as it can provide a more complete range of nutritional resources to consumers (‘balanced diet hypothesis’) (DeMott, 1998; Pfisterer, Diemer & Schmid, 2003; Worm et al., 2006). Additionally, it may enhance dietary specialization by offering greater foraging possibilities, provided that the food resources differ in nutritional quality and the high-quality food is sufficiently abundant (Araújo, Bolnick & Layman, 2011; Bolnick et al., 2011).

Despite the significance of food diversity in population dynamics, empirical studies investigating its impact on dispersal and species coexistence are largely lacking. The link between diet and dispersal has often been studied in the framework of optimal foraging theory, which broadly assumes that individuals adaptively alter their diet or behavior depending on environmental conditions (MacArthur & Pianka, 1966; Stephens & Krebs, 1986). Ideally, individuals should make fully informed dispersal decisions based on the conditions of local and distant patches (Clobert et al., 2009). Individuals may assess the quality of a patch through direct exploration or through remote detection of potential food patches by sensory cues (Bowler & Benton, 2005). Organisms may also estimate the quality of a patch based on evolved expectations of food quality and abundance (Bowler & Benton, 2005), energy intake per unit of foraging effort (Charnov, 1976), and/or contribution to fitness (Mcgraw & Caswell, 1996; Coulson et al., 2006). Nonetheless, dispersal comes with significant costs, including energy expenditure, increased risk of predation, and the possibility of reduced fitness in unfamiliar environments (Ronce, 2007; Bonte et al., 2012). These costs play a crucial role in the dispersal decisions of organisms. There is ample evidence showing that different animal species have different dispersal strategies, and these strategies can even differ between closely related species (Han & Dingemanse, 2015). Such differences in dispersal strategies are likely to evolve due to spatial and temporal variation in patch conditions (McPeek & Holt, 1992; Henriques-Silva et al., 2015). In addition, differences in dispersal behavior may play an important role in sympatric co-occurrence of closely related species (Aiken & Navarrete, 2014; Yawata et al., 2014). For instance, by dispersing to different microhabitats or utilizing different resources, closely related species that may have similar ecological requirements can minimize overlap and competition for the same resources. This strategy of niche differentiation can reduce direct competition, which may allow species to coexist within a community (Mittelbach & McGill, 2019; De Meester, Derycke & Moens, 2012; De Meester et al., 2015a). While previous studies have demonstrated that increased resource diversity may mitigate competition through enhanced niche differentiation (Martin & Garnett, 2013; Sánchez-Hernández, Gabler & Amundsen, 2017), empirical studies on the effects of food diversity on dispersal strategies, and whether these vary between closely related species and contribute to their co-existence, have rarely been carried out (Grainger & Gilbert, 2016; Kreuzinger-Janik et al., 2022).

In this regard, the cryptic species complex Litoditis marina (Sudhaus, 2011), formerly known as Rhabditis (Pellioditis) marina (Andrassy, 1983), represents an excellent candidate for testing whether food diversity can affect the dispersal behavior of sympatric closely related species. L. marina is a complex of marine bacterivorous nematodes consisting of co-occurring species that are morphologically nearly identical but genetically distinct. These species exhibit differences in ecological and functional traits, such as life histories (De Meester et al., 2015b), feeding habits (Derycke et al., 2016), and microhabitat preferences (Guden et al., 2018). They are associated with living and decomposing macroalgae in intertidal zones (Derycke et al., 2006), which are highly dynamic environments where abiotic and biotic conditions rapidly fluctuate both temporally and spatially (Moens & Vincx, 2000a, 2000b). In such heterogeneous environments, dispersal is crucial to avoid unfavourable conditions (Snyder & Chesson, 2003). Differences in dispersal strategies have been observed between the different cryptic species of L. marina (De Meester, Derycke & Moens, 2012), which can be influenced by competition (De Meester et al., 2015a). While our previous investigations revealed that food diversity can alter the life-history traits and food preferences of the cryptic species (Guden, Derycke & Moens, 2021) and can change the outcomes of intra- and interspecific interactions (Guden, Derycke & Moens, 2021, 2024), the impact of food diversity on active dispersal of L. marina remains to be tested.

In the present study, we investigated the dispersal behavior (i.e., time of nematode dispersal and proportion of dispersers) of the three cryptic species of L. marina (Pm I, Pm III and Pm IV) in patches with different food treatments (Escherichia coli, low-diversity, medium-diversity, and high-diversity food). First, we tested the effects of food diversity on dispersal in homogeneous patches consisting of the same bacterial food treatments in local and distant patches. Traditional models predict that differences in dispersal behavior between species will be largely dependent on differences in life-history characteristics in a homogeneous landscape where environmental conditions are relatively uniform across the entire area (Amarasekare & Possingham, 2001; Logue et al., 2011). We hypothesized that the differences in active dispersal between the cryptic species of L. marina are not solely explained by differences in their life-history attributes, but also by food diversity. Hence, we expected that the dispersal behavior of L. marina would also vary depending on food diversity. Second, we tested the effects of food diversity on dispersal in heterogeneous patches consisting of E. coli in the local patches and different levels of food diversity in the distant patches. Since L. marina generally exhibit higher fitness on a more diverse food source (Guden, Derycke & Moens, 2021), we hypothesized that a more diverse food in distant patches would trigger faster nematode dispersal. Investigating the effects of food diversity on the dispersal behavior of cryptic species can shed light on the evolution of feeding adaptations in closely related species, and is needed to deepen our understanding of community dynamics and species coexistence.

Methods

Nematode cultures

Monospecific cultures of three cryptic species of Litoditis marina (Pm I, Pm III, and Pm IV) were initially established from individual gravid females collected in the field. Pm I and Pm III originated from the Paulina salt marsh in the Schelde Estuary, The Netherlands (51° 21′N, 3° 49′E), while Pm IV was obtained from Lake Grevelingen, a brackish-water lake in The Netherlands (51° 44′N, 3° 57′E). The nematodes were allowed to reproduce for multiple generations to create stock cultures, which were maintained in the dark under standardized conditions (salinity of 25, temperature of 20 °C). Nematodes used in the experiments were harvested from the nematode stock cultures in exponential growth phase.

Food sources for nematodes

To investigate the effects of food diversity on active dispersal behavior of the different cryptic species of L. marina, we selected and prepared bacterial food sources according to the protocol described by Guden, Derycke & Moens (2021). Briefly, twenty-five marine bacterial strains were selected as bacterial food sources for the nematodes because they were among the most abundant bacterial taxa found in the microbiome sensu lato of specimens of at least one cryptic species of L. marina collected from the field (Derycke et al., 2016). After preparing monospecific liquid cultures of these twenty-five marine bacterial strains, the bacterial suspensions of each bacterial strain were diluted to approximately 3×109 cells ml–1, a density known to support active population growth and good individual fitness of L. marina (dos Santos et al., 2008; De Meester et al., 2011; Guden, Derycke & Moens, 2021).

Our experiments consisted of three resource-diversity treatments: ‘low-diversity food’, ‘medium-diversity food, and ‘high-diversity food’. These treatments were prepared by mixing culture aliquots of bacterial strains, as described by Guden, Derycke & Moens (2021). Low-, medium- and high-diversity food treatments consisted of five, 10, and 15 bacterial strains, respectively. For each food-diversity treatment, the component bacteria were mixed to attain approximately equal cell numbers of all bacterial strains, with a total density of approximately 3×109 cells ml–1. Because we were interested in investigating the effects of food diversity on the dispersal of the cryptic species, the replicates in each food-diversity treatment consisted of bacteria that were chosen at random from our pool of twenty-five bacterial strains. Frozen-and-thawed E. coli (strain K12, density of ca. 3×109 cells ml–1) was used as a control food treatment, which is a suitable food source that has been used in previous culture experiments with L. marina (Moens & Vincx, 1998; dos Santos et al., 2008; De Meester et al., 2011).

Experimental design

A summary of the experimental design is presented in Fig. 1. The time of first effective dispersal, defined as the dispersal followed by emergence of active juveniles in the distant patch, regardless of whether the individual was already gravid before the dispersal event, was used to measure the dispersal rates of L. marina (De Meester, Derycke & Moens, 2012). The focus on effective dispersal (i.e., dispersal followed by reproduction) ensures that the dispersal event is not solely based on the transient movement but also on the successful establishment and reproduction in the new patch. The time of first effective dispersal is henceforth referred to as ‘time of dispersal’, and implies that the longer the time of dispersal, the slower the dispersal rate of the nematodes. To investigate the effects of food diversity on the time of dispersal of the cryptic species of L. marina, we used specially designed dispersal plates, consistent with those used in previous dispersal studies on L. marina (De Meester, Derycke & Moens, 2012; De Meester et al., 2015a). These plates consist of two Petri plates (referred to as ‘local’ and ‘distant’ patches), each 5 cm in diameter, connected by a tube that is 1 cm in diameter and 10 cm in length (Fig. 1A). The substratum in the plates was provided as 60 ml of a 1.5% bacto agar medium prepared with ASW (salinity of 25; pH of 7.5–8), with additional cholesterol (100 μl–1) as a source of sterols. The agar medium was spread to a perfectly plain level across both plates and connecting tube.

Figure 1 Summary of the experimental design for testing the effects of food diversity on nematode dispersal.

(A) Specially designed dispersal plates and (B) four food-(bacteria) diversity treatments were used in the experiments. The effects of food diversity on the dispersal behavior of L. marina were tested in homogeneous and heterogeneous patches. In (C) homogeneous patches, local and distant patches were added with the same food treatments. In (D) heterogeneous patches, local patches were supplemented with E. coli and distant patches with different food-diversity treatments to investigate whether food diversity can drive dispersal.

Using the different food-diversity treatments (Fig. 1B), we performed two experimental setups with homogeneous and heterogeneous food patches to assess the effects of food diversity on active dispersal behavior of L. marina. Homogeneous patches consisted of the same bacterial treatments both in the local and distant patches. Heterogeneous patches consisted of E. coli as a bacterial food treatment in the local patch, and with a diverse food treatment in the distant patch to test whether food diversity can trigger dispersal. All three cryptic species of L. marina were used in the ‘heterogeneous’ setup, whereas only Pm I and Pm III were used in the ‘homogeneous’ set-up due to time constraints since all the experiments had to be started at the same time. In addition, Pm I and Pm III showed distinct dispersal abilities based on previous investigation (De Meester, Derycke & Moens, 2012; De Meester et al., 2015a), which make them particularly interesting in testing whether resource diversity influences active dispersal.

To investigate the effects of food diversity on the dispersal behavior of L. marina (Pm I and Pm III) in homogeneous patches, 50- μl suspensions of the same specific bacterial strains were added both in the local and distant patches. Hence, in the homogeneous-patch treatment, both patches had identical bacterial compositions. The E-E, L-L, M-M, and H-H treatments consisted of E. coli, low, medium and high-diversity food in both patches, respectively (Fig. 1C). The treatment with E. coli in local and distant patches (‘E-E’) was used as a control.

To test whether the diversity of food in an unoccupied distant patch can affect the dispersal of the cryptic species of L. marina, we assessed differences in the time of dispersal in the setup with heterogeneous patches. Here, we investigated the effects of food diversity on the dispersal of the cryptic species (Pm I, Pm III and Pm IV) by adding 50- μl suspensions of a single strain resource E. coli (strain K12) in local patches, and an equal amount of bacterial suspensions of low-, medium-, or high-diversity food in the distant patches (henceforth referred to as ‘E-L’, ‘E-M’, and ‘E-H’ treatments, respectively) (Fig. 1D). The treatment with E. coli in both patches (‘E-E’) was also used here as a control.

After adding food to the patches, five adult males and five adult females of a single species were manually picked from the stock cultures and transferred randomly to the local patches at the beginning of the experiment. No nematodes were added to the distant patches. Food was replenished in both the local and distant patches every 5 days to minimize changes to the food-diversity gradient. Using the same preparation of bacterial mixture as food for nematodes, our previous experiments demonstrated a food-diversity effect on the population growth of L. marina within a week (Guden, Derycke & Moens, 2021, 2024), supporting the persistence of diversity gradients in the food treatments for at least five days. All microcosms were incubated in the dark at a constant temperature of 20 °C, with four independent replicates per food treatment for each cryptic species. The numbers of nematodes in the distant patch were counted daily at the same time each day to check for dispersers. The timing of the arrival of the nematode(s) in the distant patches was recorded as the ‘time of dispersal’ when it was followed by reproduction (i.e., time of first effective dispersal). In all setups, the numbers of adult nematodes in the local patches at the time of dispersal were also counted to determine the proportion of nematodes that dispersed. The proportion of dispersers was calculated by counting the number of adult nematodes in the distant patch and dividing it by the total number of adult nematodes both in the local and distant patches at the time of dispersal. The experiment was conducted over a period of 20 days, during which the time of first effective dispersal was observed in all plates.

Data analyses

Differences in time of dispersal and proportion of dispersers between different food treatments were tested in R (R Core Team, 2024). All analyses were conducted with the data of adult nematodes only since it was not feasible to differentiate real juvenile dispersers and offspring of dispersed adults. Parametric tests (ANOVA) were used in all our analyses due to their robustness and effectiveness in comparing means across multiple groups. Prior to testing ANOVA, normality and/or homoscedasticity were checked using the Shapiro-Wilk test and Levene’s test, respectively. Because we were interested in investigating the effects of food diversity for each species individually and identify species-specific patterns, we conducted separate one-way ANOVA tests for each species. Posterior pairwise comparisons were performed using a post-hoc Tukey Honest Significant Differences (HSD) test.

Food diversity effects on nematode dispersal in homogeneous patches

To test for differences in dispersal rates between food treatments in homogeneous patches, a one-way ANOVA test was performed on the time of dispersal as the dependent variable and with food diversity (four levels: E-E, L-L, M-M, and H-H) as the fixed factor for each cryptic species (Pm I and Pm III). A one-way ANOVA, with the same fixed factor, was also performed to test for differences in the proportion of nematode dispersers. To test whether the proportion of dispersers was correlated with the time of dispersal, Pearson correlation coefficients were calculated for each species, and p-values were corrected for multiple testing with the Bonferroni method.

Food diversity effects on nematode dispersal in heterogeneous patches

To test whether food diversity can trigger the dispersal of the cryptic species of L. marina, we assessed differences in the time of dispersal in the setup with heterogeneous patches. A one-way ANOVA test was performed on the time of dispersal as the dependent variable and with food diversity (four levels: E-E, E-L, E-M, and E-H) as the fixed factor for each cryptic species (Pm I, Pm III and Pm IV). A separate one-way ANOVA test was also performed to test for differences in the proportion of nematode dispersers, with the same fixed factor. Pearson correlation coefficients were calculated for each species to test whether the proportion of dispersers was correlated with the time of dispersal, with Bonferroni correction for p-values.

Results

Food diversity effects on nematode dispersal in homogeneous patches

In homogeneous patches with the same food treatments in local and distant patches, food diversity showed a significant effect on the time of dispersal of Pm I (all p<0.0001) and Pm III (p<0.005) (Table 1; Fig. 2). Among all food treatments, Pm I showed the fastest dispersal in the treatment with high-diversity food (H-H), while it exhibited the slowest dispersal with E. coli (E-E) (all p<0.05) (Table 2). In Pm III, we observed significantly slower dispersal in the treatment with low-diversity food (L-L) compared to E. coli (E-E) (p<0.05) and high-diversity food (H-H) (p<0.005), but no significant differences were found between other food treatments (all p>0.05).

Table 1 Results of the one-way ANOVA tests on the effects of food treatment on time of nematode dispersal and proportion of nematode dispersers for each cryptic species of L. marina in homogeneous and heteregeneous patches. Significant differences ( p < 0.05) are highlighted in bold.

I. Time of nematode dispersal (Days)	
Species	df	F	p-value	
(A) Effect of food treatment in homogeneous patches	
Pm I	3	291.3	<0.0001	
Residuals	9			
Pm III	3	9.0	0.004	
Residuals	9			
(B) Effect of food treatment in heterogeneous patches	
Pm I	3	161.5	<0.0001	
Residuals	12			
Pm III	3	11.8	0.0007	
Residuals	12			
Pm IV	3	31.0	<0.0001	
Residuals	12			
II. Proportion of nematode dispersers	
Species	df	F	p-value	
(A) Effect of food treatment in homogeneous patches	
Pm I	3	39.3	<0.0001	
Residuals	9			
Pm III	3	29.8	<0.0001	
Residuals	9			
(B) Effect of food treatment in heterogeneous patches	
Pm I	3	17.5	0.0001	
Residuals	12			
Pm III	3	10.9	0.001	
Residuals	12			
Pm IV	3	8.0	0.003	
Residuals	12			

Figure 2 Effects of food diversity on the dispersal of the cryptic species of L. marina (Pm I and Pm III) in the setup consisting of homogeneous patches with the same food treatments in local and distant patches.

The barplot shows the (A) time of nematode dispersal and (B) proportion of nematode dispersers (mean ± SE). Food treatments in both patches consist of E. coli (E-E), low-diversity food (L-L), medium-diversity food (M-M), or high-diversity food (H-H) ( n=4).

Table 2 Pairwise-test results on the differences in time of nematode dispersal between food treatments for each cryptic species of L. marina in homogeneous and heteregeneous patches. Significant differences ( p < 0.05) are highlighted in bold.

	Time of nematode dispersal	
	Pm I	Pm III	Pm IV	
Food treatment	p-value	p-value	p-value	
(A) Differences between food treatments in homogeneous patches	
E-E vs. L-L	<0.0001	0.04		
E-E vs. M-M	<0.0001	0.9	
E-E vs. H-H	<0.0001	0.2		
L-L vs. M-M	0.02	0.08		
L-L vs. H-H	0.01	0.003		
M-M vs .H-H	0.0001	0.2		
(B) Differences between food treatments in heterogeneous patches	
E-E vs. E-L	<0.0001	0.0006	0.007	
E-E vs. E-M	<0.0001	0.2	<0.0001	
E-E vs. E-H	<0.0001	0.7	<0.0001	
E-L vs. E-M	0.8	0.02	0.02	
E-L vs. E-H	0.002	0.004	0.003	
E-M vs. E-H	0.007	0.7	0.7	

Food diversity also had a significant effect on the proportion of nematodes that dispersed at the time of dispersal for Pm I (p<0.0001) and Pm III (p<0.005) in homogeneous patches (Table 1; Fig. 2). Pm I showed higher proportion of dispersers in the two most diverse food treatments (M-M and H-H) compared to the other treatments, while the lowest proportion of dispersing nematodes was observed when both local and distant patches had E. coli (all p<0.01) (Table 3). In contrast, Pm III had the highest proportion of dispersers in the treatment with low-diversity food (L-L), and had significantly lower proportion of dispersers in the treatment with E. coli (E-E) and medium-diversity food (M-M) compared to other treatments (all p<0.05). Pearson’s correlation analysis revealed that the proportion of nematodes that dispersed was significantly negatively correlated with the time of dispersal of Pm I (Pearson’s correlation = −0.80, p<0.001) in homogeneous patches, but no significant correlation was found in Pm III ( p>0.05).

Table 3 Pairwise-test results on the differences in proportion of nematode dispersers between food treatments for each cryptic species of L. marina in homogeneous and heterogeneous patches. Significant differences ( p < 0.05) are highlighted in bold.

	Proportion of Nematode Dispersers	
	Pm I	Pm III	Pm IV	
Food treatment	p-value	p-value	p-value	
(A) Differences between food treatments in homogeneous patches	
E-E vs. L-L	0.06	<0.0001		
E-E vs. M-M	0.0001	0.9		
E-E vs. H-H	<0.0001	0.02		
L-L vs. M-M	0.008	<0.0001		
L-L vs. H-H	0.0007	0.01		
M-M vs. H-H	0.3	0.01		
(B) Differences between food treatments in heterogeneous patches	
E-E vs. E-L	0.08	0.02	0.9	
E-E vs. E-M	0.02	0.3	0.01	
E-E vs. E-H	<0.0001	0.7	0.03	
E-L vs. E-M	0.9	0.001	0.02	
E-L vs. E-H	0.003	0.003	0.04	
E-M vs. E-H	0.01	0.9	0.9	

Food diversity effects on nematode dispersal in heterogeneous patches

In heterogeneous patches with single-strain resource E. coli in the local patches and different food-diversity treatments in the distant patches, we observed a significant effect of food diversity on the time of dispersal of Pm I, Pm III and Pm IV (all p<0.001) (Table 1; Fig. 3). Pm I exhibited the fastest dispersal toward distant patches with high-diversity food (E-H) among all food treatments (all p<0.05), and the slowest dispersal toward distant patches with E. coli (E-E) (all p<0.0001) (Table 2). In contrast, Pm III exhibited the slowest dispersal toward distant patches with low-diversity food (E-L) (all p<0.05), but no significant differences in dispersal rates were observed between other food treatments (all p>0.05). In Pm IV, faster dispersal was observed toward distant patches with medium- and high-diversity food compared to other treatments (all p<0.05), and the slowest dispersal toward distant patches with E. coli (all p<0.01).

Figure 3 Effects of food diversity on the dispersal of the cryptic species of L. marina (Pm I, Pm III and Pm IV) in the setup consisting of heterogeneous patches with E. coli in the local patches and different food treatments in the distant patches.

The barplot shows the (A) time of nematode dispersal and (B) proportion of nematode dispersers (mean ± SE). Food treatments consist of a single strain resource E. coli in local patches and E. coli (E-E), low-diversity food (E-L), medium-diversity food (E-M), or high-diversity food (E-H) in the distant patches ( n=4).

In addition, food diversity had a significant effect on the proportion of nematode dispersers for Pm I (p<0.0001), Pm III (p<0.001) and Pm IV (p<0.001) in heterogeneous patches (Table 1; Fig. 3). Pm I showed the highest proportion of dispersers toward high-diversity food (E-H) among all food treatments, and the lowest proportion of dispersers toward E. coli (all p<0.05) (Table 3). Pm III exhibited significantly lower proportion of dispersers toward distant patches with low-diversity food (all p<0.05), but no significant differences were found between other treatments (all p>0.05). In Pm IV, we observed significantly higher proportion of dispersers toward distant patches with medium- and high-diversity food compared to other treatments, and the lowest proportion of dispersers toward distant patches with E. coli (all p<0.05). Pearson’s correlation analysis revealed that the proportion of nematodes that dispersed was significantly negatively correlated with the time of dispersal of Pm I (Pearson’s correlation = −0.78, p<0.001), Pm III (Pearson’s correlation = −0.59, p<0.05), and Pm IV (Pearson’s correlation = −0.73, p<0.005) in heterogeneous patches.

Discussion

Although dispersal is known to play an important role in species coexistence, the impact of food diversity on the dispersal behavior of co-occurring closely related species remains largely unexplored. Using the cryptic nematode species complex Litoditis marina as a model system, the present study provides empirical evidence that food diversity can alter the dispersal behavior of cryptic species. A summary of the results can be found in Table 4.

Table 4 Summary of the results of dispersal rates and proportion of dispersers for each cryptic species of L. marina in homogeneous and heterogeneous patches.

Species	Dispersal rate	Proportion of dispersers	
Homogeneous patches: ‘similar patch conditions’	
Pm I	H-H > L-L > M-M > E-E	H-H, M-M > L-L, E-E	
Pm III	H-H, E-E, M-M > L-L	L-L > H-H > E-E, M-M	
Pm IV	Not tested	Not tested	
Heterogeneous patches: ‘different distant-patch conditions’	
Pm I	E-H > E-M > E-L > E-E	E-H > E-M, E-L > E-E	
Pm III	E-E, E-H, E-M > E-L	E-M, E-H, E-E > E-L	
Pm IV	E-H, E-M > E-L > E-E	E-M, E-H > E-L, E-E	
Note:

(E): E. coli; (L): low-diversity food; (M): medium-diversity food; (H): high-diversity food; (>): significantly faster dispersal rate or significantly higher proportion of dispersers; (,): no significant difference between food treatments.

In a homogeneous landscape where environmental conditions (e.g., habitat quality, resource availability, and climate) are relatively uniform across the entire area, traditional ecological models predict that differences in dispersal behavior will be largely dependent on the life-history characteristics of species (Amarasekare & Possingham, 2001; Logue et al., 2011). In our experiment with homogeneous conditions where E. coli was used as the food source in both local and distant patches (E-E), we observed distinct dispersal patterns between the cryptic species of L. marina. Specifically, Pm I exhibited slow dispersal, taking approximately 15 days, while Pm III dispersed much faster, taking approximately 4 days (Fig. 2), consistent with the findings of De Meester, Derycke & Moens (2012). Pm I and Pm III differ in life-history traits such as reproductive strategy, fecundity and population growth, which can potentially explain the differences in their dispersal behavior. However, the differences in life-history traits between the cryptic species of L. marina are rather subtle and can vary depending on environmental conditions (De Meester et al., 2015b; Guden, Derycke & Moens, 2021, 2024).

If the differences in active dispersal between the cryptic species of L. marina in homogeneous patches are solely explained by differences in their life-history attributes, we would expect Pm III to consistently exhibit faster dispersal than Pm I regardless of the diversity of food in the patches. However, we observed variation in nematode dispersal depending on food diversity, indicating that factors other than life-history traits are influencing dispersal decisions. Specifically, food diversity appears to play a significant role in determining how quickly and effectively these nematodes disperse. Pm I exhibited the fastest dispersal to patches with high-diversity food (H-H) among all food treatments, which coincided with the high proportion of nematode dispersers. In contrast, Pm III showed equally fast dispersal rates in homogeneous patches with E. coli (E-E), medium-diversity food (M-M), and high-diversity food (H-H). This dispersal behavior was primarily driven by food diversity rather than population density, as both Pm I and Pm III began to disperse even at low population density in the local patch at the time of dispersal (Fig. S1).

The significant impact of food diversity on the dispersal behavior of the cryptic species is underscored by our findings on nematode dispersal in a heterogeneous condition, where E. coli was used as a food source in the local patch and more diverse food was provided in the distant patch (Fig. 3). Here, both Pm I and Pm IV exhibited faster dispersal to distant patches with a more diverse food source than to distant patches with E. coli regardless of population density (Fig. S1), indicating that food diversity mainly drives the dispersal of these cryptic species. Pm IV particularly showed faster dispersal toward the two most diverse food sources among all food-diversity treatments. Pm III also dispersed equally fast toward medium- and high-diversity food, but also showed rapid dispersal toward E. coli.

These differences in dispersal behavior depending on food diversity may be linked to food preferences. Based on taxis to food, our previous study revealed that the cryptic species of L. marina exhibit distinct food preferences based on the diversity of available resources (Guden, Derycke & Moens, 2021). Specifically, Pm III and Pm IV showed a higher attraction toward more diverse food sources, aligning with our current observations on their dispersal behavior. Different bacterial mixtures may produce distinct bacterial quorum sensing signals and/or different types or concentrations of attractants (Köthe et al., 2003; Beale et al., 2006), which could explain the food preferences of L. marina (Guden, Derycke & Moens, 2021). While food diversity did not significantly influence the movement of Pm I toward food in our previous study (Guden, Derycke & Moens, 2021), the current study reveals that food diversity influences its decision to effectively disperse, i.e., whether or not they settle and successfully breed in the new patch. Dispersal consists of three interrelated components: departure (emigration), transience (exploring the surrounding habitat), and settlement (immigrating to a different patch and successfully breeding) (Clobert et al., 2009). Since food diversity did not significantly affect the taxis to food of Pm I but played an important role on dispersal, we hypothesize that this cryptic species may move randomly or feed non-optimally during the early stages of dispersal. Animals may initially move randomly to gather information about their environment before settling in a location that offers better resources. This behavior has been observed in various species, including insects and small mammals, where individuals explore multiple habitats before making a final settlement decision (Tesson & Edelaar, 2013). Additionally, animals can also exhibit non-optimal feeding behavior during the initial dispersal phase to focus on finding a suitable habitat rather than optimizing their feeding efficiency (Croteau, 2010).

In addition to their food preferences, the dispersal behavior of the three cryptic species of L. marina aligns with their fitness performance. Our previous experiments revealed that a more diverse food source enhances the fitness of Pm I and Pm IV (Guden, Derycke & Moens, 2021, 2024). Therefore, Pm I and Pm IV tend to disperse more rapidly toward patches with a diverse food source, which they prefer, and which improves their fitness. This observation is consistent with previous studies showing that animals tend to feed optimally by choosing food resources that maximize their performance (Gripenberg et al., 2010). Interestingly, while Pm III also exhibited fast dispersal toward distant patches with medium- and high-diversity food, which have been shown to increase its fitness (Guden, Derycke & Moens, 2021, 2024), it also showed fast dispersal toward patches with E. coli regardless of population density. In the lab, all cryptic species of L. marina are easily maintained on E. coli, and Pm III exhibits the highest fecundity among all the cryptic species with this food source (De Meester et al., 2015b; Guden, Derycke & Moens, 2021). The high fitness gain (e.g., high fecundity) of Pm III from feeding on E. coli may explain why Pm III exhibits faster dispersal to E. coli than the other cryptic species.

Our findings appear to corroborate the ‘balanced-diet hypothesis’, which suggests that diverse food sources can provide a more comprehensive range of essential nutrients, thereby supporting better overall health and fitness (DeMott, 1998; Pfisterer, Diemer & Schmid, 2003; Worm et al., 2006). This hypothesis is based on the idea that a variety of foods provide a broader spectrum of nutrients, which collectively fulfill the nutritional requirements of an organism more efficiently than any single food source. For instance, DeMott (1998) found that a mixed diet of different algae provided a more balanced nutritional profile for Daphnia compared to a single type of algae. Pfisterer, Diemer & Schmid (2003) also demonstrated that herbivores feeding on a variety of plant species had better growth and reproductive rates than those feeding on a single plant species. Our findings show that the cryptic species of L. marina also likely benefit from the varied nutritional content provided by diverse bacterial mixtures, which supports their fitness and reproductive success. Our results are striking because the replicates of each food-diversity treatment in our experiment consist of different combinations of bacterial strains, which underlines the importance of food diversity per se on the dispersal of L. marina. While it would have been more informative to assess the nutritional quality of individual bacterial strains, our preliminary analyses indicated that each strain could support nematode growth. Assessing the nutritional quality would have provided insights into the nutritional value of each bacterial strain, allowing us to determine which strains are most beneficial for nematode growth and fitness. However, our preliminary analyses showed that all the bacterial strains used in the study were capable of supporting nematode growth, suggesting that the observed effects of food diversity on dispersal and fitness were not due to the inability of any particular strain to sustain the nematodes, but rather due to the overall diversity and combination of nutrients provided by the different bacterial mixtures.

The variations in the dispersal behavior of the cryptic species of L. marina based on food diversity indicate that these species gather environmental information to make dispersal decisions. This contrasts with earlier spatially-explicit ecological models that assumed dispersal was uninformed and random (Patterson et al., 2008). There is now ample evidence that animals make informed dispersal, i.e., dispersal behavior is not only influenced by the internal state of species (phenotype dependence) but also by external information (condition-dependent) (Clobert et al., 2009), such as the availability of food in a patch (Aguillon & Duckworth, 2015; Fronhofer et al., 2015; Kreuzinger-Janik et al., 2022). However, there are limited studies that investigate how environmental cues, such as food diversity, influence the dispersal behavior of cryptic species (Chenuil et al., 2019; Beheregaray & Caccone, 2007). Understanding the impact of these cues on the dispersal of cryptic species helps explain their distribution patterns, which is crucial for their conservation and management. Additionally, it enhances our knowledge of the adaptive strategies that cryptic species use to survive and reproduce, which contributes to our understanding of evolutionary processes and species adaptation (Chenuil et al., 2019).

The effects of food diversity on dispersal may also play an important role on the coexistence of cryptic marine nematode species in ecological communities. Marine nematodes are known to be capable of both passive and active dispersal. They can passively disperse through water flow, wind drift, rafting on floating items, or hitchhiking on larger animals (Thiel & Gutow, 2005; Ptatscheck & Traunspurger, 2020; Buys et al., 2021). Additionally, they can actively disperse by migrating laterally through sediments or by swimming (Ullberg & Ólafsson, 2003; Schratzberger et al., 2004; Thomas & Lana, 2011). Passive and active dispersal have been documented in L. marina (Derycke et al., 2007; De Meester, Derycke & Moens, 2012; De Meester et al., 2015a; Buys et al., 2021). In the field, the co-occurring cryptic species of L. marina live on both living and decomposing macro-algae in the intertidal zone, mainly feeding on the microbial biofilms attached to the surfaces of macro-algae. These biofilms are highly variable in (micro)space and time: bacterial communities can differ between species of algae and at different temporal scales (Lachnit et al., 2011), and may also vary between two nearby algal patches or between different structures of a single algal plant (De Meester, 2016; Guden et al., 2018). This creates spatial and temporal variability in bacterial assemblage composition, which affects the food resources available for nematodes. The resource landscape for L. marina can therefore consist of food resource patches that vary in diversity, and such heterogeneity in food diversity may alter the dispersal behavior of the cryptic species. In turn, this may have important consequences for species distribution and coexistence of the cryptic species. Our previous studies revealed that food diversity affects various life-history traits of the cryptic species (Guden, Derycke & Moens, 2021) and alters the outcome of intra- and interspecific interactions (Guden, Derycke & Moens, 2024, 2021), showing that food diversity enhances niche differentiation between the cryptic species. The current study shows that the dispersal behavior of the cryptic species of L. marina also vary depending on food diversity, which suggests that they may occupy different ecological niches. By dispersing to different patches based on food diversity, these species may reduce direct competition for the same resources. This niche differentiation may facilitate coexistence by utilizing different parts of the resource spectrum. Additionally, the cryptic species may partition resources spatially and temporally, further facilitating their coexistence. Although various factors influence species coexistence, our study provides support that food diversity may play an important role on the coexistence of closely related species (Hutchinson, 1959; MacArthur, 1965).

While the present study has expanded our knowledge on the effects of food diversity on the dispersal of cryptic nematode species, it is important to recognize certain methodological limitations. First, the results are derived from experiments conducted in controlled laboratory conditions using microcosms with a limited number of replicates, and as such may not fully capture the dynamics occurring on a larger scale in the natural environment. Hence, caution is warranted when extrapolating our results to field conditions. Second, due to time constraints since all the experiments had to be started and monitored at the same time, we did not measure mortality rates and nematode density per day, which would have given us more information on the survival and persistence of the nematodes in different food patches. Third, we did not explicitly measure bacterial reproduction rates and competitive dynamics over the course of our experiment, which could influence the composition and availability of food resources within patches. While food was regularly replenished in both local and distant patches to minimize changes in the food-diversity gradient, we recognize that bacterial dynamics could still play a role in shaping dispersal behavior. Fourth, the cryptic species of L. marina are exposed to a much higher diversity of resources in nature and more food patches than what was utilized in our study, potentially resulting in different responses. While we cannot fully mimic the diversity of resources in the real world, our research highlights the importance of resource diversity in shaping the behavior and interactions of different cryptic species, and thus may contribute significantly to the coexistence of closely related species at a larger spatial scale in the field.

Conclusion

Elucidating the mechanisms that underlie patterns of species distribution and diversity maintenance in ecological communities is a central objective in ecology. Dispersal is one of the basic life-history strategies of organisms, which can have profound consequences for meta-population dynamics, genetic diversity, and species coexistence. Using the different cryptic species of Litoditis marina as a model system, we demonstrate the role of food diversity in shaping dispersal behavior. We reveal that the cryptic species of L. marina do not disperse randomly but make informed decisions based on the diversity of available resources. This supports the idea that dispersal is not solely a random process but can be influenced by environmental cues. The cryptic species of L. marina exhibit species-specific differences in dispersal behavior depending on food diversity, but all of them tend to disperse faster toward food patches that increase fitness. The ability of the nematodes to adapt their dispersal behavior depending on food diversity indicates an adaptive strategy to optimize fitness. This adaptability is likely crucial for survival in fluctuating environments, such as intertidal zones where the cryptic species of L. marina are found. Furthermore, the differential dispersal responses to food diversity among the cryptic species may serve as a mechanism for niche differentiation. This has significant implications for the coexistence of closely related species in dynamic environments and enhances our understanding of the role of resource heterogeneity in maintaining biodiversity in marine ecosystems.

Supplemental Information

Supplemental Information 1 Population density in the local patch at the time of first effective dispersal.

(A) The numbers of adult nematodes in the local patch at the time of first effective dispersal (mean ± SE) in the local patch for homogeneous patches with the same food treatments in local and distant patches. (B) The numbers of adult nematodes in the local patch at the time of first effective dispersal (mean ± SE) for heterogeneous patches with E. coli in the local patches and different food treatments in the distant patches (n = 4).

Supplemental Information 2 Effects of resource diversity on active dispersal behaviour of 3 cryptic species of the nematode Litoditis marina.

The legend for each data point is available in "Metadata".

Nele De Meester is gratefully acknowledged for her valuable inputs on the experimental design.

Additional Information and Declarations

Competing Interests

Author Contributions

Data Availability

The authors declare that they have no competing interests.

Rodgee Mae Guden conceived and designed the experiments, performed the experiments, analyzed the data, prepared figures and/or tables, authored or reviewed drafts of the article, and approved the final draft.

Sofie Derycke analyzed the data, authored or reviewed drafts of the article, and approved the final draft.

Tom Moens conceived and designed the experiments, authored or reviewed drafts of the article, provided funding for this research, and approved the final draft.

The following information was supplied regarding data availability:

All data of the study is available in the Integrated Marine Information System (IMIS) database (VLIZ): Guden, R.M.; Derycke, S.; Moens, T.; Marine Biology Research Group, Department of Biology (Ugent); Aquatic Environment and Quality, Flanders Research Institute for Agriculture, Fisheries and Food (ILVO): Belgium; (2023): Effects of resource diversity on active dispersal behaviour of three cryptic species of the nematode Litoditis marina. Marine Data Archive. https://doi.org/10.14284/610.

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
