# Peer review of "To stay or to go: resource diversity alters the dispersal behavior of sympatric cryptic marine nematodes"

_PeerJ, doi:10.7717/peerj.18790_

## Round 0.1 · original submission · Major Revisions

This manuscript examines dispersal from one petri dish to another in three species of marine nematodes in relation to the absolute and relative species diversity of its bacterial food. Three distinct experiments were carried out (but see below for possible overlap). There were highly significant differences among species and treatments, but it was not easy to grasp any overall pattern.

As the reviewers point out, this is an interesting study, written in excellent English, but incomplete, making it impossible to evaluate.

I started a detailed review, but became frustrated by the incompleteness and difficulty of understanding the results and discussion and ran out of time.

The manuscript requires major revision, and the reviewers and I will take another, careful look at it when all the information for a critical evaluation is included. My comments reflect that I stopped a detailed reading before completing the manuscript, so you should make sure you understand the general issues and similarly revise the sections for which I did not provide detailed comments.

Manuscript formatting does not follow Instructions to Authors. There is an explicit instruction to left justify only (not full width), not to embed figures and tables in text and to refer to figures as Fig., except when starting a sentence.

In contrast to other problems, the quality of the reference section is one of the best of have seen in recent manuscripts. I only found a few errors such as a failure to italicize species names (L463) and a capital A needed in MacArthur’s name (also in text).

Note that Reviewer 2 presented their review as a pdf. It is listed as an annotated manuscript, but it is, in fact, a regular review, not an annotated manuscript.

Editor’s Comments

The abstract is very hard to understand. It isn’t clear what ‘effective dispersal’ is and how it is measured. It refers to nematode density which was not clearly a manipulated variable. It is not clear what is meant by ‘adding local patches’. It is not clear that diversity is varied by changing number of bacterial species; I read the parenthesis as if somehow diversity was varied using only E. coli, which I could not understand. Remember that in many literature searches, this is all that potential readers will see, so it must be clear, complete, and compelling, as well as concise.

As both reviewers point out, the Methods section is very incomplete. Remember that another researcher should be able to repeat the experiment with the information provided.
• The nematode species and bacterial strains are well described. Since cultures started with a single individual, is it possible that any conclusions do not apply to the species but to the related offspring of that individual?
• Introduce the abbreviations for treatments when they are first described.
• I could not find any appropriate description of the apparatus beyond Fig. 1. I would like to see a justification of the design and a clear description of the chambers and how nematodes could disperse between them. A reader should be able to construct an identical apparatus, if needed. For example, I did not see any mention of the inside diameter and material of the connecting tube and whether it filled with medium.
• The measures of dispersal are very unclear in what they mean, how they were made, and how they relate to measures used in the broader dispersal literature. Should you make sure that readers understand that higher values imply lower dispersal rates?
• I did not see any information about how often measurements were made.
• I could not see any indication of replicate number for the experiments.
• How might the densities of nematodes and bacteria change over the course of the experiment and what effect might this have on the interpretation?
• The reviewers raise some concerns about the statistical analysis. Please consult a statistician to be sure that your conclusions are valid.

I did not find it easy to grasp the results.
• Your clearest hypotheses relate to the effects of the relative and absolute diversity of the inoculated patch and the vacant patch. You predict that species of nematodes will differ but not how. Thus, it is confusing for the results to present species differences first and sometimes exclusively. Would it be clearer to explain the treatment differences and then which were consistent between species or not.
• It does not seem necessary to provide p-values in scientific notation to more than 4 decimal places. Is there any reason not to adopt the p < .05, .01, .001, and .0001 convention?
• The Methods indicated post-hoc analyses were carried out, but I did not see this in the Results. I only realized that I could find these patterns later, when I discovered Table 2, which was not cited until the Discussion. These are results and belong in the appropriate section.
• I found Table 2 very confusing because the symbols indicating greater than (>) represent not dispersal speed but latency, which is the opposite, if I understand correctly. This is not completely clear in the table heading.
• I notice that the results for experiments with EE in both locations are extremely similar for species I and III in experiment 1 and 2. Is experiment an independent replication which resulted in exactly the same result (which should be highlighted if so) or is it that you used the same data in two ‘different’ experiments (which should be clearly identified in the Methods and taken into account in the statistical analysis, if so)?
• For the effect of diversity on dispersal in homogenous patches, the Results mainly presents species comparisons. However, the question requires a within-species analysis of effects of diversity. There was a significant treatment effect in the ANOVA, but a significant interaction as well. I feel that it is completely unclear if and how diversity effects dispersal. Are the non-linear patterns in relation to diversity significant or potentially a reflection of sampling or natural variation?
• Again, for the heterogenous patches, the Results focuses on species differences and does not present the effect of diversity, which I thought was the main question. It seems particularly noteworthy that one of the three species dispersed fastest in EE while the other two were slowest in this treatment (assuming the differences are significant, which was not presented here).
• I did not carefully examine the third section of Results.

The Discussion needs to be much more thorough and to carefully justify the conclusions, including their strengths and weaknesses.
• Does your concept of push and pull reflect the use of these terms in the literature? If I understand correctly, you use ‘push’ to refer to the value of the target patch (I would have considered this a pull because dispersers are being pulled out of their original location) and you use ‘pull’ to consider the value of the inoculation patch (I would have considered this a push because dispersers are pushed to leave by poor conditions). I may have this wrong (it wouldn’t be the first time!), but you should consider the terms and if and how they have been previously used.
• The first section on effect of diversity in homogenous patches should not start with a broad literature review, which sounds more like an introduction, but instead focus on the strengths and weaknesses of evidence for a diversity effect on dispersal between similar patches. The lack of differences among many treatments and the inconsistency between species limit any decisive statements about density and dispersal. Consider both measures of dispersal and how they relate to each other. Then you can develop a discussion of different speeds of dispersal between species within treatment categories. Finally, relate these to the literature on this specific topic.
• Similar issues apply to the remaining sections.

Reviewer 1 ·

Excellent Review

This review has been rated excellent by staff (in the top 15% of reviews)
EDITOR COMMENT
This was a critical but constructive review that identified important gaps in the presentation. It was well organized and helpful for the editor and, hopefully, for the authors as well.

Basic reporting

1.1) Clear and unambiguous, professional English used throughout
This study investigates the dispersal of 3 cryptic marine nematode species in response to different resource diversity. This was obtained by creating 2-patch system (dishes) connected via a 10 cm corridor. The 2 patches were inoculated with different combinations of bacterial resources (namely E. coli only or inocula containing low, medium and high bacterial diversity). For example, patch 1 (proximal patch, starting population) and patch 2 (distal patch, dispersal population) were both inoculated with E. coli, or patch 1 with E.coli and patch 2 with low bacterial diversity and so on in a similar fashion. The dispersal of the nematodes, from patch 1 to patch2, was therefore assayed under these different food combinations. The article is well written and tackle some interesting questions, however it is currently a bit hard to evaluate the study due to some lack of information. I have some major and fundamental points that need to be addressed by the authors in all the sections of the manuscript. I will proceed following PeerJ guidelines.

1.2) Literature references, sufficient field background/context provided.
The introduction is somehow confusing and is missing some references. I would start from the general predictions in the last paragraph of the introduction (ll.98-111). For instance, why “diverse food sources in distant patches would trigger fast dispersal” (l.105)? The general overall prediction is that a decrease in local patch quality (or a decrease in spatio-temporal predictability of patch quality) would trigger and favour dispersal. The main hypothesis (and the logic of the introduction) should be refined and built from this starting point, see for example the following meta-experiment https://www.nature.com/articles/s41559-018-0686-0

or other studies on dispersal with the most famous nematode C.elegans
https://onlinelibrary.wiley.com/doi/full/10.1046/j.1461-0248.2003.00524.x
https://onlinelibrary.wiley.com/doi/full/10.1111/jeb.13563

Following on the main general prediction, a main question I had during the revision is which one is the best food for these nematode species? Did you test survival, reproduction and other key traits under different resource diversity? I think you described quickly something in the discussion this would be actually a main point. This may also help refining your basic predictions. Also, considering the nematode as a predator and the food resource as a prey, there is a whole bulk of literature on state- and context-dependent dispersal (plastic dispersal) from a predator-prey and host-parasite perspective.


1.3) Professional article structure, figures, tables. Raw data shared.
The “Methods” section is not really clear, and key information for reproducibility of the study are missing. Particularly, because of this lack of details, it was slightly difficult to follow stats and results section. Some info is present in Figure 1 but (at least) a whole paragraph should be added in the main text. The figures and tables are OK per se, but I would suggest to run the statistical analysis differently, which would imply some different figures and tables. See below the section “Experimental Design”.


1.4) Self-contained with relevant results to hypotheses.
This is fine.

Experimental design

2.1) Original primary research within Aims and Scope of the journal.
As above 1.1, the manuscript is well written and interesting but requires some major revisions.

2.2) Research question well defined, relevant & meaningful. It is stated how research fills an identified knowledge gap.
The main predictions should be better explained (see 1.2)

2.3) Rigorous investigation performed to a high technical & ethical standard.
2.4) Methods described with sufficient detail & information to replicate.
I am combining these 2 points.

Here some fundamental major questions that need to be clarified.
-How does the starting cultures in the experimental patch 1 were established?
-Did you control for initial density?
-How do many initial worms were used at the start? Yes/No implications
-Were the worms synchronized in terms of age? Yes/No implications
-Sex-biased dispersal?
-How did you take dispersal measurements? Every day, twice a day…?
-Did you check for mortality in patch 1 or patch 2? Yes/No implications, e.g. are they dispersing because of resource or it is mortality driven?
-Did you check density over the course of the experiments?
-Were the experiments run on the same period?

Because you let the worms in patch 1 for several days, a measure of density or mortality would be pretty important. How can you distinguish whether the dispersal behaviour is not a consequence of the local patch dynamic (mortality for instance)? This is fundamental problem that you need to explain or justify properly. Typically, these sorts of dispersal experiments are run over few hours to avoid this problem, or population density are constantly measured (pushed or pulled way dynamics in range expansion terminology). The density at dispersal is relatively informative, as that density comes from some previous interactions in the populations. Further, if age was not controlled, worms of different age are likely to behave and feed differentially. For the scientific solidity of the study and interpretation, it is very important to better explain these points.


-Food sources for nematodes (paragraph at l.122).
I am a bit concerned about this, again because the experiments lasted several days. How did you control/you know that 1 strain was not outcompeting the others or becoming dominant in the dish for the food diversity treatments? If this was the case, the low, medium high diversity food resources would become meaningless. Could you please develop more?

-Data analysis (paragraph l. 147)
I was pretty confused by the statistical analyses. I would recommend to do some different analyses more in line with the literature. I don’t think what you are doing is conceptually wrong but I find it pretty hard to interpret and far from the golden standard. In addition, looking at the figures, it seems to me that there is a positive correlation between time until dispersal and number of adults (something to check). Again, I am a bit worried about the potential non-standardization of density at the beginning of the experiment in patch 1.

Instead of analysing time and density separately, I would suggest to use the proportion of dispersal individuals on the total individuals (something that you have, thanks for sharing the data), with time as additional factor.
In R, the complete model will be something like this:
mod1<-glm(cbind(df$no.dispersal.plate,df$no.inoculation.plate) ~ df$treatment*df$species* df$day.dispersal , family=quasibinomial)

If the experiments were run at the same time, you could actually have one big model, but you would have to exclude the Pm IV strain.
Why this is present in only one assay? You should explain.
And why by looking at Figure 1 not all combinations of food were assayed?

An alternative option to look at the data is to have one column for the food treatment in patch 1 and food in patch 2 (again with the exclusion of the Pm IV strain).
The model would become something like
mod2<-glm(cbind(df$no.dispersal.plate,df$no.inoculation.plate) ~ df$patch1* df$patch2*df$species* df$day.dispersal , family=quasibinomial)

It is also not clear to me how did you performed the specific contrast between treatments. Did you control for multiple testing (Bonferroni correction or similar)?

I would also recommend to show these proportional data instead of using barplot (maybe some boxplot instead, or at least show the raw proportional data).
If you really think it is more interesting to use time as response variable, I would still try to standardize everything by density and have 1 single model with both time and density.

Validity of the findings

3.1) Impact and novelty not assessed. Meaningful replication encouraged where rationale & benefit to literature is clearly stated.
If the authors can justify the raised issues, the study would add a nice paper to the literature of dispersal plasticity / informed dispersal in heterogeneous environment.

3.2) All underlying data have been provided; they are robust, statistically sound, & controlled.
Data and meta-data have been provided. As described above, I would really suggest to reanalyse the data. The main effects are there but the details of the interactions and therefore some part of the discussion might change.

3.3) Conclusions are well stated, linked to original research question & limited to supporting results.
I think it is necessary to rework the introduction and the original hypotheses in light of some additional theory, and to better explain the design with all the additional information (properly address question at point 2.3-2.4). Some sentences are pretty strong and need to be toned down, for example ll. 301-302. Some additional discussion and introduction on how the worms would perceive food differences and make their choice (some mechanistic explanation) would be good. Further, the conclusion might slightly change due to the reanalysis of the data, which I recommend.

Reviewer 2 ·

Excellent Review

This review has been rated excellent by staff (in the top 15% of reviews)
EDITOR COMMENT
This was a constructively critical review that identified some very important issues and presented them in a clear, straightforward and objective way, while providing supportive comments to the authors.

Basic reporting

Please see attached pdf.

Experimental design

Please see attached pdf.

Validity of the findings

Please see attached pdf.

Additional comments

Please see attached pdf.

Annotated reviews are not available for download in order to protect the identity of reviewers who chose to remain anonymous.

---

## Round 0.2 · Major Revisions

I apologize for the delay in completing my review of your manuscript. It proved to be unusually difficult to find two suitable reviewers. Neither original reviewer was available, so I had to find new ones. I was diligent in my search but had a large number of negative responses as well as many who never answered. The first review was completed weeks before I managed to find a second reviewer. The good news is that, in the end, we ended up with two excellent, thoughtful reviews which will be very helpful to your publication.

I found that the Introduction and the manuscript in general are much clearer than the previous version and that overall it was much better focussed and more interesting to read. There are a few places where I thought minor changes in the text would improve clarity, and the reviewers noted some others. My suggestions are on an attached pdf, with highlights to indicate the problematic words or punctuations and comments to propose insertions. Note that you have not been completely consistent in the use of UK and USA spelling. I noticed both ‘behaviour’ and ‘behavior’. Please choose one language for the entire manuscript (except references, of course) and run a spell check to be sure that you are consistent.

Major Concerns

In the comments on the previous version, I suggested consulting with a statistician and you indicated in your rebuttal that you had done so. Nevertheless, Reviewer 1 indicated that ANOVA did not appear to be an appropriate approach with the type of data you have. I hope you will consult again, showing the statistician the comments, to check whether your approach is truly valid. If your approach is valid, perhaps a brief comment to justify the choice might help other readers who would have similar concerns.

Similarly, this reviewer points out that there is no formal comparison of the differences among species which would be of interest.

Reviewer 2 has valuable suggestions regarding background information on movement and reproduction rates needed to put your findings into a more ecological context. I think it is important to consider this aspect carefully so that readers can relate your study to the broader dispersal literature.

Minor Suggestions

L20-22. This is key information but not clear. I think you need to specify that bacteria were the food and specify the number of species in each treatment and clarify that the single species E. coli in both chambers was a type of control. The reader won’t know the relevance of E. coli otherwise.
L26-27. This is also unclear, partly because the sentence structure switches from nematode species first to treatment first and partly because the verb switches from speed of dispersal to the trigger for dispersal implying that they are different variables. (‘Trigger’ is the wrong word here.)
L58-59. Adaptive alteration of diet is an assumption of optimal foraging theory, not a prediction. The predictions come from applying this assumption to particular sets of choices.
L184ff. Indicate how long the experiment ran.
L226ff. Instead of just reporting significance levels, it would be useful to give an overview of the actual values of time of dispersal and proportion of population dispersing at that time.

Tables
In your rebuttal, you agreed to put your p-values into the conventional categories rather than scientific notation, but I note that you did not make these changes consistently in the tables.
Also, as suggested by Reviewer 2, there is room to put the mean values (and perhaps ranges) in the tables.

Figures
As the reviewers noted, Figs. 2 and 3 are not very clear. The captions state that they show the effects of diversity on dispersal. However, you have one of your dependent variables on the y-axis and the other on the x-axis, not the independent variable (diversity) on the x and the dependent variable on the y as would be needed to clearly show the effect. These figures seem to be the correlations between the two dependent variables but the way they are expressed with box plots for one and not the other are not clear. Figures to show the effect may not be needed if you included the actual values in the tables. On the other hand, for visual thinkers, figures make a stronger impression. Perhaps you should think about using the figures to highlight the correlations rather than show the effects of the independent variables. Be sure the caption matches what is actually presented.

Also, boxplots can have different formulations so need to be fully described in the caption (horizontal line, vertical line, box, points). If I am not mistaken, box plots are normally used for non-parametric data showing medians and interquartile ranges. I wonder if they are even relevant for sample sizes as small as four. Please check into whether your graphical approach is valid.

The figures are also missing units of time.

Reviewer 3 ·

Basic reporting

Overall I found this paper to be a reasonable test of the effect of resource diversity on dispersal. The experimental design and methodology seem reasonable, and the paper was generally easy to follow. The a-priori context for why resource diversity would matter in the first place seems reasonable, and the results are generally interpreted well in the discussion. I had wondered whether the movement recorded could actually be considered dispersal, as opposed to, for example, just foraging, but the timing of dispersal was measured when dispersing nemotodes reproduced in the distant patch, not just when they arrived.

Experimental design

The proportion of dispersers was measured at the time of first arrival in the other patch and the overall number of individuals in both plates differed among treatments. I wondered if and when ‘treatment effects’ on proportion of dispersers is confounded with population growth rate e.g. if nemotodes reproduce more or quicker in some treatments, is their dispersal a response to the food or the density?

The data are analyzed using ANOVA. The dependent variables are ‘time of dispersal’ and ‘proportion of dispersers’. The former is bounded by 0 (can only be positive) and the latter is bounded between 0 and 1. An ANOVA assumes the residuals are normally distributed (i.e., not bounded), which clearly cannot be the case with these dependent variables, especially when the time of dispersal is less than about 5, and when the proportion of dispersal is closer to 0 or 1. Ideally, the data should be modelled using generalized linear models (glm) (e.g., perhaps gamma for time of dispersal, and binomial for proportion of dispersers) and effects tested using log-likelihood ratio tests. That said, the effects seem so large here, that I would guess that the p-values would still be less than 0.05. So I think the ANOVA results are quantitatively ‘wrong’ (because they’re modelling parameter spaces that cannot exist), but are still qualitatively correct (i.e, the inferences are correct). On a related note, modelling the data properly with a glm would allow the reporting of effect sizes to tell the reader ‘what was the biologically meaningful effect size’, which is usually more interesting than ‘is there a statistically significant effect’, as well as showing plots of means and 95% confidence intervals, which is much more informative that the boxplots that are shown.

One of the main, and quite interesting, conclusions was that the dispersal behavior differed among species. However, species were never formally compared in the statistical models. A better analysis of how species differ would include ‘species’ in the model with ‘treatment’ (e.g., evaluate species*treatment and evaluate species+treatment).

I would find the presentation of Fig. 2 and 3 to be better if it also showed plots separating the response variables to always be on the y-axis, and the treatment variable to always be on the x-axis, like how the data were analyzed, to see treatment effects.

Table 1: An F-test has two degrees of freedom, but only one is shown here (numerator). The dominator degrees of freedom needs to be presented as well, which I think would be 12? This table also has A) and B) repeated.

Figure 1D: Are the orange E.coli dishes supposed to be grey in color?

Validity of the findings

The data were provided and I was able to re-create the results (for what I looked at).

The discussion provides a reasonable interpretation of the data, but it is very specific to this particular nematode system and does not really place the results into a broader context (at least not beyond a superficial level)

Reviewer 4 ·

Basic reporting

This article examines the effects of prey diversity on the movement of three closely related nematode species between an initially occupied and an unoccupied container in a laboratory setting. The study shows between-species differences in movement rates, and different effects of prey diversity on movement in some of the species. The article is generally well written, the literature referenced is appropriate, and the figures are generally good (although some possible improvements are mentioned in the 'General Comments' section

Experimental design

The experimental design is reasonable for examining the effect of prey diversity on movement. However, some details of the frequency of observation should be provided to assess the accuracy of the movement rate estimates.

Validity of the findings

The general findings that prey diversity in the occupied and unoccupied patches both affect movement rates is valid. However, the result is not surprising, and the biological implications of this result are not very clear.

Additional comments

This section is divided into specific (4A) and general comments (4B). The specific ones are generally issues of wording and clarity in presentation. The more substantive issues are discussed in the general comments section. These call for some changes in the results and discussion sections.

4.A. Specific

p. 1- Abstract I found the description of food treatments hard to understand from the abstract. It would be good to state here whether the more diverse food treatments included E. coli or not. This becomes clear in the main text, but should be clear from the abstract. Saying ‘medium and high diversity’ would be clearer than ‘the two most diverse’. Why would E. coli trigger dispersal when the control patch had E. coli? I am left wondering if Pm III dispersed more slowly towards a patch with low diversity alternative food than it did towards a patch with just E. coli (which represents a lower diversity than the ‘low diversity’ treatment.
Line 20 – ‘had’ would be better than ‘with’
Line 36 – ‘an informed process’ is somewhat unclear; ‘dependent on available information about conditions in the current patch and alternative patches’ would be clearer.
Lines 31-44 - It would be good to mention that remote detection of conditions in other patches can influence dispersal. In addition, there are usually costs to dispersal, which influence the movement decision.
Line 46 – ‘has been’ would be better than ‘remains’, because the reference provided is 22 years old.
Line 54-55 – Having more diverse foods will only enhance dietary specialization if the foods differ in nutritional quality and the high quality foods are sufficiently abundant. Toxins or diseases contracted by eating (among other things) could also influence the effects of food diversity if these negative factors differ between food types.
Line 63 – Delete ‘olfactory and visual’, since these may not be the only possible sensory cues
Line 67 - It is unclear if ‘animals’ refers to one or more species—given the second half of the sentence, I think you mean ‘different animal species’
Line 81 – No references are given here, but the word ‘rarely’ does suggest that there have been some previous studies.
Line 85 - This would be clearer if you ended the sentence after ‘distinct’ and began a second sentence substituting ‘These species’ for ‘which’
Line 98-99 – ‘proportion of dispersers’ was unclear when I first read this. It implies that, at the time that dispersal is first detected, there may be more than one disperser. However, this seems suggest that the presence in the initially unoccupied patch may not have been monitored sufficiently often. (I am assuming nematodes do not have coordinated movements?). This also raises the question of whether an individual could visit the unoccupied patch and return to the initial patch without this being detected.
Line 99 – It would be good to give a reference for or at least explain the names of the cryptic species (‘Pm’ followed by a Roman Numeral)
Line 106 – Delete ‘only’- Using ‘solely’ makes this redundant.
Line 115 – ‘of’ is better than ‘on’
Line 123 – Does the salinity figure need some sort of units?
Line 142 – ‘total’ would be better than ‘final’ (if this is what you mean)
Line 146 – Does freezing kill the E. coli? (I assume it does) If so, is it necessary to say something about decomposition rate?
Line 171 – It might be good to say whether De Meester et al. used a similar petri-dish+tube setup as in the present study.
Line 175 – It is not clear from the text whether the homogeneous patch treatments involved having the same specific bacterial strains in both patches. The notation seems to allow for L-L to mean the 5 same strains in each patch OR simply the same number (5) of randomly chosen strains in each patch.
Line 190 – I think it is important to be clear about the timing of the dispersal observation-are the observations made once a day at the same time each day?
Line 216 - It would be clearer if you stated that you are talking about whether the diversity of food in an unoccupied patch can affect dispersal.
Line 277 - The papers cited here discuss dispersal between similar patches on a long time scale. Here it appears that the dispersal is more of a case of foraging movements in a system in which occupied patches can be rapidly depleted. (However, I may be wrong about this—thus it would be good to know something about timing of movement relative to lifespan of the nematodes.)
Lines 277-78 - I think the statement about dispersal in a uniform environment being dependent on life-history characteristics is somewhat misleading. Many other factors, including risk of inbreeding without or with low dispersal, are important, and could easily be more important than life history traits.
Line 295 - I would have liked to see more explanation of food preferences, including some information about what Guden et al 2021 and 2024 found regarding food preferences.
Line 313 – 14 - Optimality should be some balance between the performance of the parents and the offspring—not just offspring.
Line 333 – There are many models of dispersal that assume nonrandom and/or informed movements – I personally have at least 5 publications assuming informed adaptive movement—they may not qualify as spatially explicit, since most contain a small number of patches. This shouldn’t be a limitation since the experiments described here have only two patches.
Line 337 – I do not see why being part of a cryptic species complex would cause a species to have less informed or more informed dispersal strategies than any other species that is not part of a cryptic complex.
Line 362 – That food diversity may contribute to coexistence was one of the main messages of Hutchinson’s and MacArthur’s works on competition in the 1960s—at least one of their earlier papers should at least be referenced here. While the nematode species may be harder to distinguish visually than are birds, it is not clear that they are any less different in ecologically important characters.
Line 377 – It might be good to specify that you are talking about coexistence ‘at a larger spatial scale’
Line 380 - Remove the comma after ‘communities’.
Fig. 2 legend. I assume the dots are the individual replicates. However, with n = 4 it is strange to see only two or three dots. I assume this is because one or more dots are covered up by the box that has the same color—I would recommend just having a colored outline of the box. In any case, define n and define what the box is on the legend. In addition, there needs to be a (M-M) following ‘medium-diversity food’.
Tables: I would recommend redoing Tables 1-3 to include the estimated dispersal times or rates as well as p values.

4.B. General and more major comments

(1) It seems important to know something about the suitability of the different bacteria for the different nematode ‘species’. There is a reference to the previous 2021 and 2024 studies by these authors showing that food identity has effects on fitness that differ between species. However, there is no indication in this paper of whether the effect on fitness was taken into account in choosing prey species in the experiments reported in this manuscript.
(2) The definition of ‘time of first dispersal’ (line 150) is somewhat unusual because it includes both dispersal AND reproduction in the new patch. I think this choice of definition should be justified. It would also be good to know the time of gestation. It is also important to discuss what (If anything) is known or assumed about the possibility of remote detection of food presence/abundance in the initially unoccupied patch (see specific comment regarding Lines 197-99). In any case, I think the measure of dispersal needs more explanation and justification. There seems to be no consideration of homeward movement following dispersal, which should be a possibility. It would also be nice to know how long it was between arrival and reproduction. There also appears to be an assumption that the bacteria do not move between the during the experiments; maybe this should be documented or at least discussed.
It would also be good to know whether your unusual definition of dispersal (to include reproduction) affects the measure greatly. In other words, how much would the dispersal time be changed by not including reproduction. In any case, it would be good to know the movement rate of the nematodes and the frequency of observations to determine whether remote assessment of conditions the other patch is involved and whether an unobserved round trips between home and away patches would be possible between observations.
(3) I found the presentation of results somewhat frustrating. The important underlying question for me is what biological processes underly the dispersal. It is not surprising that diversity of food would have some effect when the food types known differ in quality and the total amount is similar. This creates an immediate connection between diversity of prey and the rate at which the mean quality of the prey in a patch should decline, as a result of the most preferred food types being exploited first. There are other possible mechanisms, including competition between or differences in life history of the bacteria. Reproduction of bacteria and interactions between the bacteria in the patches would affect their abundance over time, so there needs to be some comment on whether these processes occur during the experiments.
(4) It seems clear that food diversity per se does not explain dispersal. If it did, there should be no cases in which the 5-prey treatment was no more attractive (or even less attractive, as in fig 3B) than the E.coli treatment. The important open question (at least for me) is, what is causing the differences in movement rates in treatments with different prey diversities. There are basically two reasons for an organism to be attracted to a nearby area with a greater diversity of food types. One is that consuming a variety of types increases fitness relative to consuming single type. (because of containing different toxins or nutrients). This is the ‘balanced diet’ mechanism. The second is that the foods simply differ in the amount of a single desired nutrient. In this case, the most diverse set of food organisms is most likely to have the highest quality foods, but in the current design, those foods may not be very abundant in the higher diversity treatments. Movement takes place mainly to get the highest quality food. The outcome depends on how fast these high-quality foods are depleted. The cited article by Derycke et al (2016) evidently showed differential resource use and different microbiomes for the species in this complex. It would have been nice to know more about what this earlier paper showed as a possible explanation for understanding the effects of the diversity of foods on fitness.
(5) The paper ends with a statement that implies the authors are ultimately interested in what maintains diversity of nematode species. Competition may or may not be the only or the most important factor which limits nematode diversity in natural systems (predation could be involved). It would have been good to say how the movement differences between the different L. marina types could affect their coexistence.
(6) I think that most readers will be frustrated by the lack of analysis/discussion of why diversity affects movement. It looks like the Pm I treatment with HH had roughly half of the individuals disperse (which includes reproducing) on day 1. This is a huge difference when compared to the EE treatment. Is it possible that the high diversity treatment has some toxic bacteria?

---

## Round 0.3 · accepted · Accept

The authors have made appropriate changes in response to the reviews and have provided a detailed description of their responses. The manuscript is now much clearer and more readable. I consider it ready for publication.